# Parenting beliefs and psychological distress → stimulation and punishment → young children's behavior: A descriptive study in Colombia

Jorge Cuartas [1,2,3*], Catalina Rey-Guerra [1,3], Juliana Borbón [1,3]

1 Department of Applied Psychology, New York University, United States of America, 2 Centro de Estudios sobre Seguridad y Drogas (CESED), Universidad de los Andes, Colombia, 3 Fundación Apapacho, Colombia

* j.cuartas@nyu.edu

## Abstract

Parental engagement in stimulating activities and the use of different discipline behaviors play a significant role in young children's behavioral development. Psychological frameworks often posit that parental beliefs and psychological well-being are key drivers of these parental behaviors. However, the influence of parental beliefs and psychological well-being on these parenting behaviors, and consequently on children's behavioral outcomes, remains understudied, particularly in low- and middle-income countries (LMICs). We collected primary data from 267 parents of young children ($M_{age}$ in months = 13.66; 52.06% girls) living in low-income households in Bogotá and Soacha, Colombia, and assessed how parental beliefs and psychological distress predict children's behaviors through parental engagement in stimulation activities and use of violent punishment. Structural equation modeling indicated that positive parental beliefs about violence were linked to reduced engagement in stimulating activities, predicting lower child effortful control and positive affectivity, while parental psychological distress was associated with greater use of violent punishment, predicting lower effortful control and higher negative affectivity in children. These results underscore the importance of addressing parental beliefs and psychological well-being to support positive parenting behaviors and promote healthy behavioral development in young children, particularly in LMIC contexts.

## Introduction

Promoting positive parenting behaviors has gained substantial momentum in global research, policy, and programmatic work aimed at fostering young children's development [e.g., 1–3]. Significant progress in children's health and nutrition has shifted the global policy focus from ensuring that young children "survive" to also helping them "thrive" [3]. This policy shift has also been motivated by evidence showing that less than 30% of children younger than five years living in low- and- middle-income

**Data availability statement:** The data is available in the OSF, as described in our manuscript.

**Funding:** Grants from the PanAmerican Health Association and Parenting for Lifelong Health.

countries (LMICs) receive adequate early learning opportunities like stimulation and protection from violence like violent punishment in their homes [4], which developmental scholars have long considered as "proximal processes" and critical engines of early development [e.g., 5].

One critical question in research and practice is why some parents provide high levels of stimulation (e.g., engagement in play and other learning activities) and restraint from using violent punishment early in their children's lives, while others do not, and how to promote positive parenting more broadly. A central hypothesis in psychological science is that parenting cognitions (e.g., beliefs and attitudes) and emotions (e.g., psychological distress) shape behaviors such as engagement in stimulating activities and punishment, thereby indirectly influencing child development and behavior [6–8]. Despite this, empirical work testing the simultaneous role of cognitions and emotions in influencing parenting behaviors and child outcomes is scarce, particularly in LMICs that have been traditionally underrepresented in developmental science [9–12].

Furthermore, current intervention research and practice aimed at promoting stimulation and preventing violence in the home has primarily focused on shifting cognitions, including beliefs and attitudes about violence, while overlooking the role of emotions such as supporting parental psychological well-being [e.g., 13–15]. Beliefs and attitudes that legitimize violence are still prevalent and the psychological well-being of parents has deteriorated due to the COVID pandemic and growing humanitarian emergencies, particularly in multi-burdened LMICs [e.g., 16]. Therefore, it remains essential to understand the way beliefs and psychological well-being influence the parenting behaviors that can help children thrive.

The main purpose of this descriptive study is to examine the associations between parenting beliefs and attitudes toward violence and psychological distress, stimulation and punishment, and young children's behavior using primary data collected from a sample of low-income families in Bogotá and Soacha, Colombia. Colombia is an upper-middle-income country characterized by high levels of contextual adversity due to a protracted civil conflict and violence in the home [17]. The country is making significant investments in the prevention of violence and the promotion of positive parenting as a means to promote resilience and children´s development, including the National Strategy to Prevent Physical Punishment [18] and the National Guidelines to Preventing Violence in Early Childhood [19]. These recent policy tools strongly emphasize shifting beliefs and attitudes to promote positive parenting but only mention parents' psychological distress marginally. Within this context, a better understanding of how early beliefs and psychological distress influence parents' stimulation, punishment, and child outcomes early in life is important to informing and improving existing and future policy and programmatic efforts.

## Conceptual framework

We employ a bioecological perspective as an overarching framework to understand parenting behaviors and child development and behavior in context [5,20,21]. This perspective includes the following key features: (1) culture and key characteristics of

the meso-, exo-, macro-, and chronosystems are intertwined with each other and within the immediate microsystem; (2) these features probabilistically influence the nature of parenting and human development, which in turn influence distal and proximal contexts (i.e., multicausality and simultaneous causality or reciprocity); and (3) distal factors influence parenting and human development via more proximal factors. An ecological perspective on parenting suggests that (a) parent characteristics (e.g., beliefs and emotions), (b) child characteristics (e.g., age and gender), and (c) the household and family environment (e.g., parent education, employment, household composition, and wealth) interact to shape parenting and child development [22]. At the same time, more distal factors such as parental employment or household wealth likely exert their influence primarily through proximal predictors, including parents' cognitions and emotions [8,21,23].

In this study, we focus on proximal predictors of parenting, namely *beliefs and attitudes* and *psychological distress*, and on *stimulation* and *violent punishment* as two significant parenting behavior that shape *child development and behavior*. We understand *beliefs* as cognitions that parents may take as true and acquired from their culture, experiences, and education and are entwined with *attitudes* or predispositions and evaluations of beliefs [8,24]. In turn, we conceptualize *psychological distress* as an umbrella term that includes painful mental and somatic symptoms, such as feeling depressed, restless, fearful, and having difficulties concentrating [25]. Moreover, we define *stimulation* as the cognitively, socially, and emotionally enriching activities and interactions that parents engage in with children in or around the home, which have also been conceptualized as parental investments in the economics literature [26] and playful parenting [27]. Finally, we understand *violent punishment* as physical and psychological violence used to control or punish children's behaviors, which can include spanking, hitting children with objects, and using humiliation or guilt [28].

Below, we discuss how beliefs and psychological distress may influence stimulation and punishment, ultimately affecting child development and behavior.

## Beliefs → stimulation and punishment

Beliefs and attitudes (hereafter beliefs) can affect behaviors by providing frameworks for assessing the potential value or utility of different behaviors, guiding interactions [29]. Cross-cultural evidence has demonstrated that about one out of three parents in LMICs believe in the instrumental value of physical punishment or the idea that children need to be physically punished for being raised properly [30]. Beliefs about the instrumental value of physical punishment, as well as widespread beliefs about physical punishment being harmless and a defining part of being a "good parent" [31], can lead to a narrative where using violent punishment is necessary and positive, therefore increasing the likelihood of its use. At the same time, the connections between beliefs and behaviors may be content-specific, meaning that beliefs about physical punishment will likely influence the use of physical punishment but not necessarily of other parenting behaviors, e.g., stimulation [8].

While evidence from LMICs is still nascent, prior findings resonate with the idea that parents' beliefs affect parenting behaviors. For example, one study of 2,254 primary caregivers of young children in Ghana showed associations between endorsement of physical punishment, measured using a single yes or no item on believing that physical punishment is needed to bring up, raise, or educate a child properly, and the likelihood of using physical punishment and psychological aggression to punish children [32]. Similarly, one cross-national study included more than 180,000 caregivers of young children in 65 countries and used the same single-item measure of beliefs about physical punishment, and found associations between believing physical punishment is necessary and a 155% increase in the odds of the children being spanked [33]. Yet, more research is needed using more comprehensive measures of beliefs (instead of a single item) and assessing the links between beliefs and behaviors in other cultural contexts.

## Psychological distress → stimulation and punishment

Emotions are now also considered critical drivers of behavior, in addition to beliefs and cognitions. As parenting is a demanding task that requires both cognitive (e.g., knowledge and adequate beliefs) and emotional (e.g., self-regulation

and patience) resources, appropriate emotional regulation and activation is necessary for avoiding harmful parenting behaviors while ensuring responsive, sensitive behaviors [6,34]. Indeed, psychological distress may make it harder for parents to identify children's cues and deploy the cognitive and emotional resources needed to respond to their needs and behaviors [35]. Therefore, psychological distress can increase the risk for low engagement in learning activities with children, as well as avoidant or violent responses that can be more consequential early in life when children are highly dependent on their caregivers [36,37].

Examining the links between psychological distress and parenting behaviors is particularly relevant to LMICs, where contextual adversity is prevalent. Empirical evidence from Bangladesh, Brazil, India, Uruguay, and South Africa, among other LMICs, suggests that parental psychological distress is associated with lower engagement in stimulating activities and a higher likelihood of resorting to violent punishment [38]. Contextual violence and poverty-related hardships that exacerbate psychological distress tend to be more prevalent in LMICs relative to high-income countries [39], which is the case in Colombia where more than half a decade of civil conflict has triggered widespread psychological problems in victims and those "indirectly" exposed to the conflict [40].

### Stimulation and punishment → child behaviors

Parental engagement in stimulating activities has often been linked to positive child outcomes both in high-income and LMICs, but stimulation looks quantitatively and qualitatively different across cultures and settings. For instance, studies using large-scale cross-sectional data from 44 countries have shown associations between parental engagement in reading, playing, singing, and telling stories (among others) and positive developmental outcomes [41]. Other studies using longitudinal data from pilot studies in five LMICs also show links between stimulation and young children's social, emotional, and executive function skills [42]. At the same time, data from nationally representative surveys demonstrate that parents' engagement in stimulating activities with young children tends to be lower, on average, in LMICs relative to high-income countries and households with lower wealth relative to wealthier households [43]. Ethnographic evidence also shows how the same activity (e.g., playing) can manifest in different ways across settings due to differences in societal beliefs, socialization goals, and type of activities and skills that are valued [e.g., 44, 45].

Similarly, violent punishment can increase the risk for negative child outcomes, specifically behavioral outcomes, but there is still discussion on whether the effects of violent punishment may vary according to cultural factors. Attachment [46] and social learning theories [47] indicate that violent punishment can erode the parent-child relationship and model violence, therefore increasing the risk for the onset of child behavior problems. Neurodevelopmental perspectives posit that exposure to violence may make children felt threaten, which can shift neural circuits in ways that increase hypervigilance and aggressiveness [48]. While existing evidence from meta-analyses and reviews of longitudinal studies show robust associations between violent punishment and negative behavioral outcomes [49,50], there is still discussion on whether these effects may vary according to beliefs and how normative is violent punishment within a culture [51]. Therefore, it remains important to examine the associations between stimulation, violent punishment, and child behavioral outcomes in more cultural settings to assess specificity vs. generalizability in observed associations.

### Psychological distress → stimulation and punishment → child behaviors?

To summarize, there is evidence for pairwise relations (a) beliefs → behaviors; (b) psychological distress → behaviors; (c) stimulation and punishment → child outcomes. Yet, there is less examination of how beliefs and psychological distress, simultaneously and independently, may influence child outcomes indirectly by shifting parenting behaviors (with some exceptions from high-income countries, e.g., [9]). Moreover, samples from LMICs have been vastly underrepresented in related research, and significant variation in beliefs, psychological distress, and parenting behaviors across cultures and settings demonstrate the importance of examining these developmental processes in diverse settings and cultural contexts.

### The present study

This study aims to examine the beliefs and psychological distress → stimulation and punishment → child behavioral outcomes model presented in Fig 1 using primary data from a sample of low-income families in Bogotá and Soacha. Specifically, we seek to respond to the following research questions:

- RQ1: What do low-income parents report as their beliefs about violent punishment and psychological distress, engagement in stimulating activities with children, and use of violent punishment in Bogotá and Soacha, Colombia?

- RQ2: Does parents' beliefs and attitudes towards violent punishment, as measured by agreement to four statements, and parents' reports on nine symptoms of psychological distress represent unified latent constructs of beliefs and psychological distress, respectively, for the Colombian context?

- RQ3: Are parents' beliefs and psychological distress associated with stimulation and violent punishment and, in turn, children's behavioral outcomes?

## Methods

### Positionality, ethical approval, and transparency

We begin by presenting a positionality statement, following recent suggestions on increasing developmental science's generalizability and relevance [12] and acknowledging that our identities and lived experiences may inform methodological decisions and interpretation of the data. The three authors of the current study are Colombian. The first author self-identifies as male. He is a Spanish-speaking scholar based at a U.S. academic institution who grew up in a low-income household in Bogotá. The second author identifies as a cisgender woman and scholar who uses her research to amplify marginalized voices from the Global South and challenge dominant narratives within academic discourse. The third author identifies as a cisgender woman, researcher, and has lived her entire life in Colombia.

We conceptualized and collected data for this study between June and October 2024 in partnership with Innovations for Poverty Action (IPA) and the Instituto Colombiano de Bienestar Familiar (ICBF), which is the public institute in charge of child and family policy in Colombia. The study received ethical approval from the Institutional Review Board (IRB) at IPA on June 28, 2024. The data, measurement instruments, and analytic code for this study are available through our project's Open Science Framework repository at https://osf.io/xv9ma/?view_only=0a439c7a76b648418ea5a727c3bc7c63

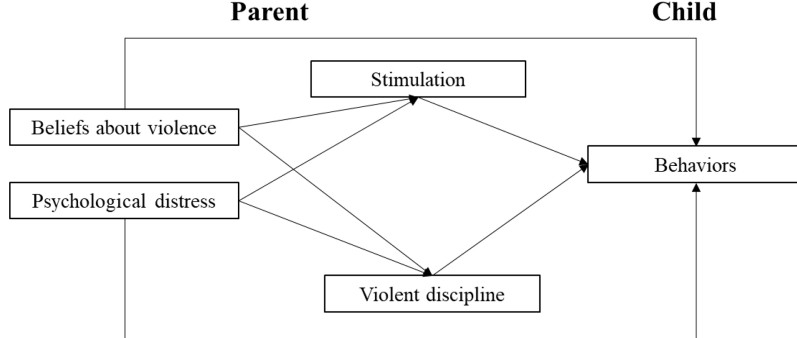

**Fig 1. Conceptual model.**

## Procedures and sample

This study uses primary data collected during the pilot of a violence prevention parenting program. The research team, IPA, and the ICBF identified the localities of Mártires and San Cristóbal in Bogotá, as well as Soacha, as focal areas due to significant increases in domestic violence reported between 2022 and 2023. Our focus was on families receiving services from ICBF through the *Modalidad Familiar* (Family Modality or home-based care). Families attending *Modalidad Familiar* come, in general, from low-income households who are eligible for social services according to the Unified Vulnerability Assessment and Identification System for Social Assistance – SISBEN. We held meetings with coordinators of all Modalidad Familiar Service Units (UDS) in the targeted areas to identify which units could participate based on availability. Seventeen UDS were selected, and all caregivers in these units with children under five were eligible and invited to provide written informed consent. In total, 353 individuals from these units provided informed consent to participate in the study. The recruitment period took place between 03/07/2024 and 17/07/2024.

IPA led the data collection between July 17 and 25, working with 14 trained enumerators who surveyed 267 parents using a survey instrument in Spanish. Enumerators identified the primary caregiver of the children by asking for the person who attends most of the group meetings in *Modalidad Familiar.* They also identified a focal child for data collection by asking the primary caregiver to respond to all questions based on one of the children younger than five under their care (when more than one). The sample comprised 265 mothers (99.25%) and 2 fathers, aged 28.94 years old on average (*range* $= 18 - 59$), 177 (66.29%) had secondary or less as their maximum education level and 90 (33.71%) post-secondary education, 77 (28.84%) had a job, whereas 190 (71.16%) claimed they did not work and had an average of 1.16 children younger than five under their care in their household (*range* $= 1 - 5$). Focal children were, on average, 13.66 months old (*range* $= 1.2 - 57.6$139 (52.08%) were girls, and 128 (47.94%) were boys. Finally, 199 (74.81%) respondents stated that the focal child's father lived in their home, whereas the father lived in another household for the other 67 (25.19%). (See Table 1 for further details).

## Measures

*Beliefs.* We developed a 4-item scale to assess parents' beliefs and attitudes towards violence, based on prior qualitative work [52] and related studies [e.g., 30]. Specifically, parents responded their level of agreement to the following four statements on their attitudes towards violent punishment (item b) and beliefs about its instrumental value (items a, c, and d) using a 4-point Likert scale (*1 = disagree; 2 = partially disagree; 3 = partially agree, 4 = agree*): (a) spanking children when they misbehave teaches them how to behave; (b) a good father or mother spanks or slaps their child when they misbehave; (c) when a child is hit, they will stop misbehaving; and (d) yelling at a child makes them more obedient (authors' translation). We calculated an average score of beliefs, with higher scores representing more positive beliefs or attitudes towards violent punishment ($\alpha = .80$).

*Psychological distress.* We employed a modified version of the CES-D-10 to assess psychological distress. The CES-D-10 is a 10-item Likert scale questionnaire (*1 = never; 2 = once; 3 = sometimes, 4 = all of the time*) for the frequency of depressive and somatic symptoms, which been used in the Colombian context [e.g., 53]. We removed the item *I could not "get going"* because the ICBF did not approve its use in populations receiving their services, stating that it was misaligned with their policy framework. Therefore, we used a 9-item version of the CES-D-10. We reversed the coding of two items on positive affect and computed a mean score, with higher scores representing more psychological distress ($\alpha = .75$).

*Stimulation.* Parents reported the frequency (*1 = never; 2 = a couple of days; 3 = most days, 4 = all days*) mothers, fathers, and other adult caregivers engaged in the following eight activities in the last week, following the Family Care Indicators [54]: (a) read books or look at pictures in books together; (b) tell stories; (c) sing songs or sing together, including lullabies; (d) take the child outside the house, for example, to the market or to visit relatives; (e) play any simple game together; (f) name objects or draw things together; (g) show or teach something new, such as a new word; (h) counting

**Table 1. Descriptive Statisticss.**

| Variable | N | M or % | SD | Min | Max |
|---|---|---|---|---|---|
| Beliefs | 267 | 1.37 | 0.42 | 1 | 2.5 |
| Psychological distress | 267 | 1.88 | 0.5 | 1 | 3.33 |
| Stimulation | 267 | 2.99 | 0.51 | 1.38 | 4 |
| Violent punishment | 267 | 1.16 | 0.26 | 1 | 2.71 |
| Negative affect | 257 | 2.59 | 0.71 | 1.22 | 4.67 |
| Positive affect | 257 | 3.71 | 0.61 | 1.4 | 5 |
| Effortful control | 257 | 4.08 | 0.45 | 2.7 | 5 |
| Child age (months) | 266 | 13.66 | 8.8 | 1.2 | 57.6 |
| Child gender | (N = 267) | | | | |
| Male | 128 | 47.94% | | | |
| Female | 139 | 52.06% | | | |
| Parent age (years) | 266 | 28.96 | 6.78 | 18 | 59 |
| Parent highest education level | (N = 267) | | | | |
| Primary or Secondary | 177 | 66.29% | | | |
| Superior | 90 | 33.71% | | | |
| Parent works | (N = 267) | | | | |
| Yes | 77 | 28.84% | | | |
| No | 190 | 71.16% | | | |
| Father at home | (N = 266) | | | | |
| Yes | 199 | 74.81% | | | |
| No | 67 | 25.19% | | | |
| Wealth | 267 | −0.01 | 0.98 | −2.05 | 2.47 |
| Overcrowding | 266 | 1.96 | 0.76 | 0.43 | 6 |
| Number of children under 5 | 267 | 1.16 | 0.47 | 1 | 5 |

games (e.g., involving addition or subtraction) or teach numbers. We computed a mean score with higher scores representing more engagement in play and other learning activities with the child ($\alpha = .68$).

*Violent punishment.* Parents reported how often (*1 = never; 2 = some days; 3 = most days, 4 = always*) they and other adult caregivers called the child by names, ignored for a prolonged time, threatened with abandoning or hitting him or her, yelled, shaken, spanked, or hit the child with objects, to "discipline" or punish the child. We computed a mean score with higher scores representing more frequent use of violent punishment ($\alpha = .60$).

*Child behavior.* We used the Infant and Early Childhood Behavior Questionnaires [55] to assess positive affectivity, negative affectivity, and effortful control or regulatory capacity. Prior evidence links these behavioral outcomes with the emergence of internalizing and externalizing behavior problems in childhood and adolescence [e.g., 56]. The positive affectivity domain includes 12 items such as "how often during the last week did the baby smile or laugh when given a toy?" and "when hair was washed, how often did the baby vocalize?" and demonstrate adequate reliability in the current sample ($\alpha = 0.73$). The negative affectivity is comprised of 12 items like "at the end of an exciting day, how often did your baby become tearful?" and "when the baby wanted something, how often did s/he become upset when s/he could not get what s/he wanted?" ($\alpha = 0.83$). Finally, the effortful control included 12 items like "when singing or talking to your baby, how often did s/he soothe immediately?" and "when being held, in the last week, did your baby seem to enjoy him/herself?" ($\alpha = 0.60$).

*Covariates.* Finally, we collected data on children's age in months and gender, parents' highest education level, whether the parent works, and an indicator for whether the father of the child lives at home. We also gathered information about

the number of children younger than five and the number of rooms to sleep and household members, which we used to compute an overcrowding index. Parents also reported whether they possessed 12 different assets, including a bicycle, car, laptop, and access to the internet in their homes, among others, which we used with a principal component analysis to compute a household wealth index [57]. The variables child's age, parents' age, overcrowding, and whether the father lives at home had one missing value each, which we replaced by the average value for each variable.

## Analytical approach

For the present confirmatory study, we used two analytic strategies. First, we conducted a descriptive analysis to assess parents' beliefs about violent punishment, psychological distress, engagement in stimulating activities with children, and use of violent punishment in Bogotá and Soacha, Colombia (RQ1). Second, we used Structural Equation Modeling (SEM) to examine the direct and indirect associations between parents' beliefs about violence, psychological distress, and early childhood behavioral outcomes via stimulation and punishment (see conceptual model in Fig 1). SEM allowed us to: (a) model both observed and latent variables while accounting for measurement error [58]; (b) simultaneously test multiple mediating processes; and (c) compare different model specifications concerning hypothesized relationships between predictors, mediators, and outcomes [59].

We began by testing the measurement model for the two latent variables: beliefs and psychological distress (RQ2). We assessed convergent validity, or the extent to which the items are correlated with each other, therefore indicating they may be measuring the same underlying construct. Then, we tested two structural models to identify the best-fitting model: (a) a fully mediated model excluding direct paths from parents' beliefs and psychological distress to child behavioral outcomes, and (b) a partially mediated model including the paths that were excluded in the fully mediated model (RQ3). We included covariances between latent variables and error terms of endogenous variables. We evaluated model fit using standard criteria: comparative fit index>.90 (CFI), Root Mean Square Error of Approximation (RMSEA) <.05, and Standardized Root Mean Square Residual (SRMR) <.08 [60]. Given that the two models were nested, we also compared their relative model fit using a log-likelihood difference test [61]. We included child age, child sex, parents' age, parent education, parent working status, an indicator for whether the father of the child lives at home, the number of children younger than five living in the household, an overcrowding index, and a wealth index as covariates predicting all mediators and outcome variables. We used bias-corrected bootstrapping with 1,000 draws to generate accurate confidence intervals to test the strength of indirect paths between predictors and outcome variables via each of our hypothesized mediators [58]. We conducted all analyses in Mplus Version 8.3 [62] using a Maximum Likelihood with Robust Standard Errors estimation (MLR) approach and adjusting standard errors for the sampling design (i.e., families clustered within UDS) with the CLUSTER command.

## Results

### RQ1: Variation in beliefs, psychological distress, and parenting behaviors in bogotá and soacha

Tables 1 and 2 present descriptive statistics and correlations for the study variables, and Fig 2 shows the proportion of parents endorsing each item in the beliefs item (top) and psychosocial distress index (bottom). On average, parents reported strong disagreement or disagreement with the four statements endorsing violence (M = 1.37; SD = 0.42), although more than 10% agreed with the instrumental value of physical punishment, that is, the belief that spanking teaches children to behave. Most parents reported experiencing at least one symptom of distress (M = 1.88, SD = 0.50), with the most common being feeling that everything requires effort (66.67%), feeling hopeless about the future (62.88%), and feeling afraid (54.14%). Regarding parental behaviors, the data indicated relatively high engagement in stimulating activities (M = 2.99 out of 4, SD = 0.51), alongside a prevalent use of violent punishment (M = 1.16 out of 4, SD = 0.26; See S1 and S2 Fig in the supplemental file).

Table 2. *Bivariate correlation matrix.*

| Variables | 1 | 2 | 3 | 4 | 5 | 6 | 7 | 8 | 9 | 10 | 11 | 12 | 13 | 14 | 15 |
|---|---|---|---|---|---|---|---|---|---|---|---|---|---|---|---|
| (1) Beliefs | | | | | | | | | | | | | | | |
| (2) Psychological distress | .07 | | | | | | | | | | | | | | |
| (3) Stimulation | −.20** | −.09 | | | | | | | | | | | | | |
| (4) Violent punishment | .20** | .34*** | −.12+ | | | | | | | | | | | | |
| (5) Negative affect | −.04 | .29*** | .04 | .19** | | | | | | | | | | | |
| (6) Positive affect | −.03 | −.06 | .37*** | .08 | .25*** | | | | | | | | | | |
| (7) Effortful control | −.15* | −.20** | .26*** | −.25*** | .02 | .39*** | | | | | | | | | |
| (8) Child age (months) | .06 | .15* | .10 | .46*** | −.05 | .23*** | −.26*** | | | | | | | | |
| (9) Child gender | 0.01 | −0.05 | 0.04 | 0.04 | −0.06 | 0.09 | 0.04 | 0.08 | | | | | | | |
| (10) Parent age (years) | .13* | .04 | .00 | .10 | −.03 | .00 | .01 | .10+ | −.01 | | | | | | |
| (11) Parent highest education level | −.05 | −.18** | .11+ | −.06 | −.09 | .07 | .02 | .04 | .08 | .12+ | | | | | |
| (12) Parent works | −.11+ | −.04 | .02 | .10 | .01 | −.01 | .06 | .05 | −.06 | .09 | .11+ | | | | |
| (13) Father at home | .02 | −.09 | .07 | −.03 | .00 | .03 | −.02 | .02 | −.09 | −.07 | .03 | −.09 | | | |
| (14) Wealth | −.09 | −.14* | .16** | −.02 | .01 | .02 | .05 | .04 | .01 | .09 | .27*** | .16* | .10 | | |
| (15) Overcrowding | −.06 | .07 | −.06 | .01 | .01 | −.10 | .03 | −.08 | .01 | −.09 | −.17** | −.10 | .08 | −.18** | |
| (16) Number of children under 5 | .01 | .12+ | −.15* | .21** | .00 | −.01 | .01 | .47*** | .01 | .13* | −.11+ | .10 | −.04 | .04 | .15* |

*Note.* ***p < .001, **p < .01, *p < 0.05, +p < .10.

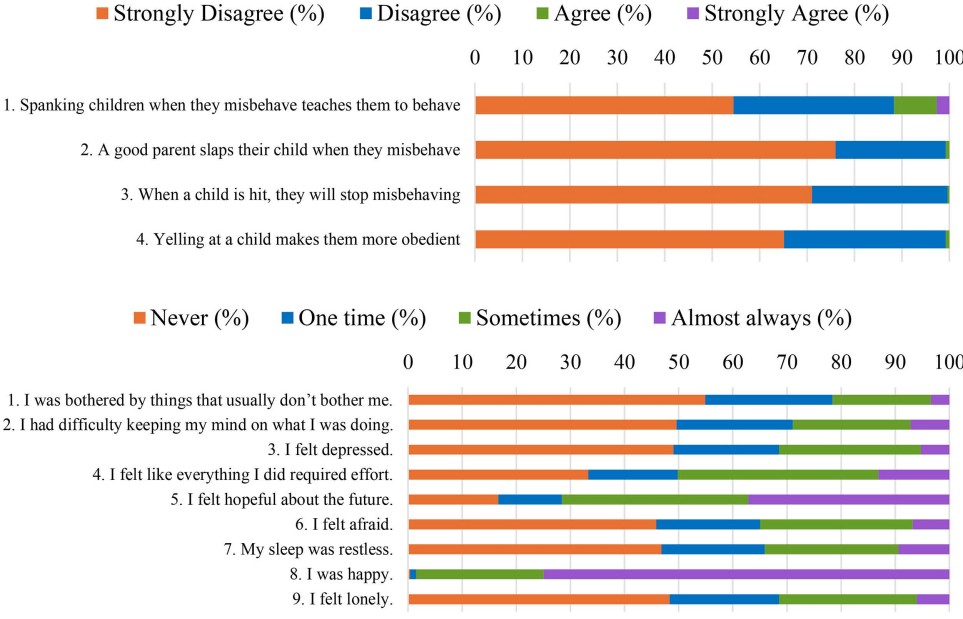

**Fig 2. Parents' beliefs about violence (top) and reports about psychological distress (bottom).**

## RQ2: Measurement of beliefs and psychological distress

Results from the measurement model indicated adequate convergent validity for the two latent variables, with standardized loadings > .40 (see Fig 3). The measurement model including the two latent variables demonstrated adequate fit overall, $\chi^2(78) = 675.4$; $p < .01$; RMSEA = .04; CFI > .95; SRMR = .05. The factor structure and overall fit of the measurement

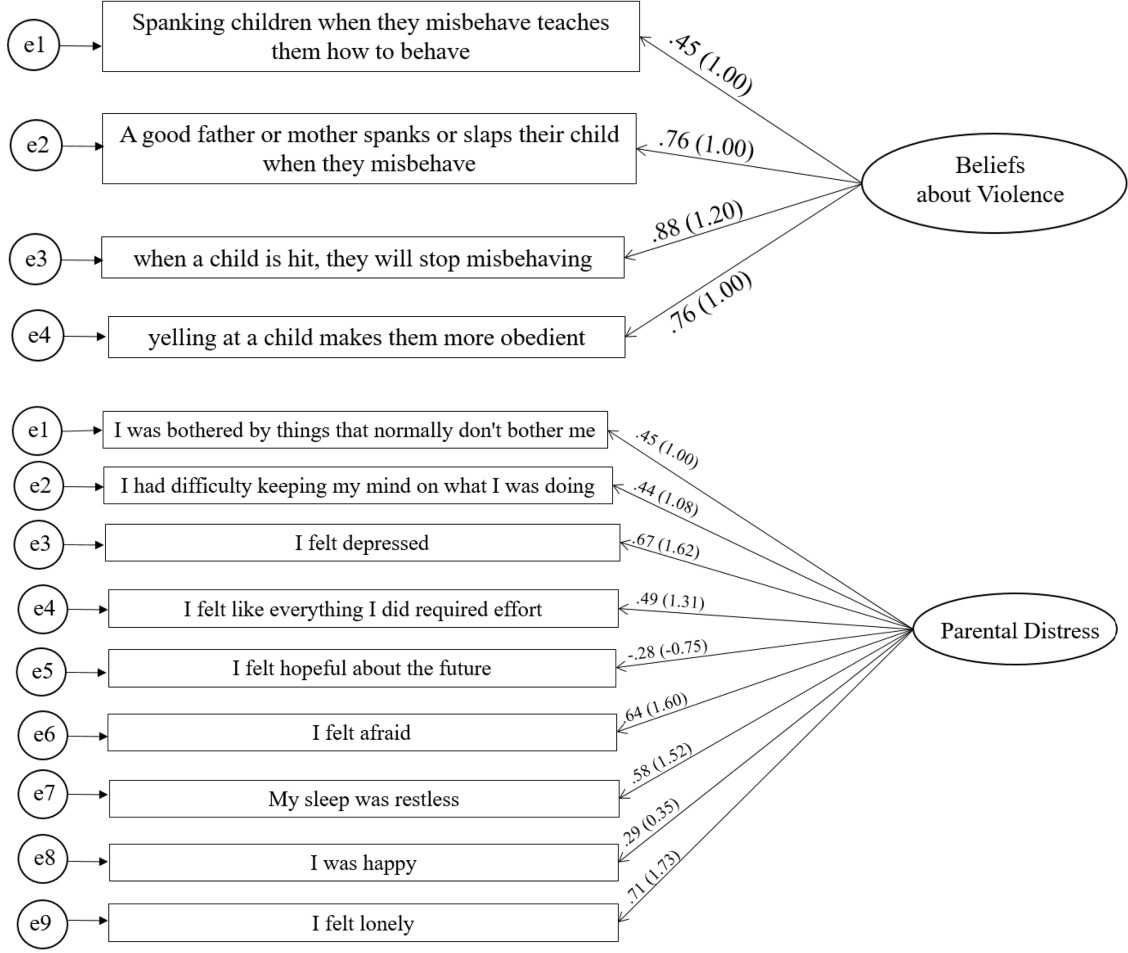

**Fig 3. Standardized Factor Loadings for the Latent Variables.** *Note.* Standardized coefficients (and unstandardized on parenthesis) are presented. All coefficients are statistically significant at $\alpha < 0.001$. Maximum likelihood estimator with standard robust errors to non-normality and clustered at the USD level. Model fit for beliefs about violence: $\chi2(2) = 3.12$; RMSEA = 0.05; CFI = 0.99; TLI = 0.98, and model fit for psychological distress: $\chi2(27) = 20.14$; RMSEA = 0.04; CFI = 0.96; TLI = 0.94.

model were adequate, allowing us to proceed with the structural model. Fig 3 presents the standardized and unstandardized factor loadings from the CFA for beliefs about violent punishment and psychological distress.

### RQ3: Associations between beliefs, psychological distress, stimulation, violent punishment, and child behavior

The partially mediated model shown in Fig 1 (hereafter, Model 1) demonstrated adequate fit (RMSEA = 0.04; CFI = 0.91; SRMR = 0.05). In comparison, the fully mediated alternative (Model 2), which excluded the direct paths from parents' beliefs to psychological distress and child behavior outcomes, showed slightly poorer fit (RMSEA = 0.04; CFI = 0.90; SRMR = 0.05). To formally compare these nested models, we conducted a log-likelihood difference test, evaluating whether adding direct paths in Model 1 significantly improved model fit. The test indicated a significant improvement, $\Delta\chi^2(4) = 18.30$, $p = .001$, favoring Model 1. Thus, we retained Model 1 as the final analytical model.

## Direct paths

Table 3 presents unstandardized coefficients (*b*) with standard errors, *p*-values, and standardized coefficients (*β*) for all direct paths in Model 1. To aid interpretation, Fig 4 presents all direct effects with p-values < .10 along with their standardized coefficients, visually highlighting the key mediating processes identified in the SEM model. We did not find any statistically significant direct associations between parental beliefs about violence and child outcomes. Results indicated a positive association between psychological distress and children's negative affectivity (*β* = .32, SE = .07, p < .001), but no significant associations with effortful control or positive affectivity. Stimulation was positively associated with both effortful control (*β* = .33, SE = .08, p < .001) and positive affectivity (*β* = .35, SE = .07, p < .001), but not with negative affectivity. The path from violent punishment to effortful control was marginally significant and negative (*β* = −.12, SE = .06, p < .10), whereas the path to negative affectivity was marginally significant and positive (*β* = .15, SE = .08, p < .10). No significant association was found between violent punishment and positive affectivity.

## Indirect paths

Several indirect paths emerged, suggesting that parental stimulation and violent punishment may partially account for the associations between parental beliefs, distress, and child behavior outcomes (see Table 4). Specifically, beliefs about violence were indirectly associated with lower effortful control (β = −.060, 95% CI [−.148, −.010]) and lower positive affectivity (β = −.063, 95% CI [−.144, −.020]) through reduced stimulation. Parents' psychological distress was indirectly associated with lower effortful control via increased violent punishment (β = −.036, 95% CI [−.084, −.004]) and with higher negative affectivity via violent punishment (β = .045, 95% CI [.001, .092]). No other indirect paths reached statistical significance.

**Table 3.** *Unstandardized and Standardized Coefficients for Direct Effects.*

| Path | *b* | SE | *p*-value | β |
|---|---|---|---|---|
| **Effortful control** | | | | |
| Stimulation → | .308 | .077 | .000 | .330 |
| Violent punishment → | −.210 | .110 | .056 | −.116 |
| Beliefs about violence → | −.083 | .130 | .523 | −.060 |
| Psychological distress → | −.102 | .106 | .333 | −.088 |
| **Positive affectivity** | | | | |
| Stimulation → | .423 | .081 | .000 | .347 |
| Violent punishment → | .245 | .185 | .185 | .104 |
| Beliefs about violence → | .008 | .093 | .935 | .004 |
| Psychological distress → | −.058 | .096 | .545 | −.038 |
| **Negative affectivity** | | | | |
| Stimulation → | .123 | .113 | .278 | .087 |
| Violent punishment → | .396 | .229 | .083 | .145 |
| Beliefs about violence → | −.249 | .179 | .165 | −.119 |
| Psychological distress → | 0.568 | .144 | .000 | .323 |
| **Stimulation** | | | | |
| Beliefs about violence → | −.268 | .122 | .028 | −.180 |
| Psychological distress → | −.103 | .089 | .249 | −.083 |
| **Violent punishment** | | | | |
| Beliefs about violence → | .035 | .064 | .582 | .046 |
| Psychological distress → | .202 | .039 | .000 | .314 |

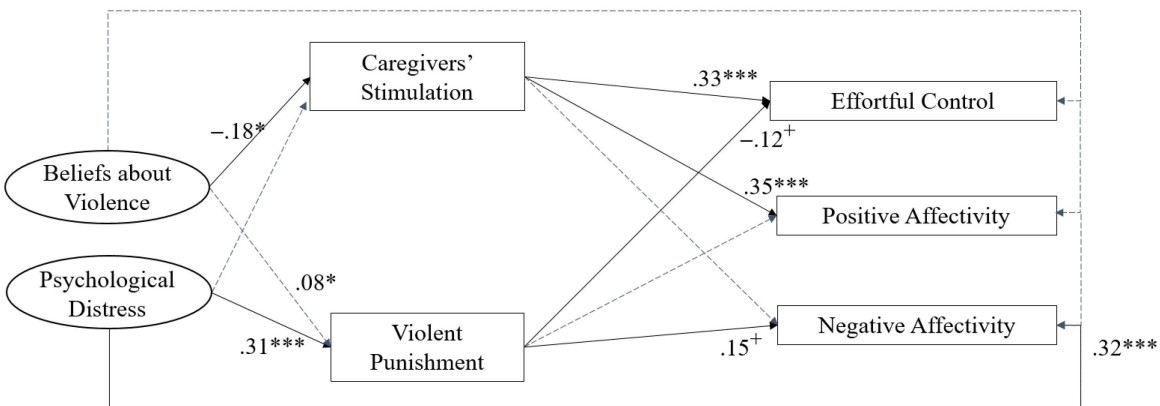

**Fig 4. Standardized Coefficients of Direct Paths Between Key Study Variables.** *Note. N* = 267. Standardized coefficients are presented. Maximum likelihood estimator with standard robust errors to non-normality and clustered at the municipality level. All exogenous variables are correlated. Solid black lines represent statistically significant direct effect pathways, whereas the dotted gray lines represent direct effect pathways that did not reach statistical significance. Table 3 presents full results, including unstandardized coefficients and standard errors. RMSEA = 0.04; CFI = 0.91; SRMR = 0.05. ***p < .001, **p < .01, +p < .10.

**Table 4. *Standardized Coefficients for Indirect Effects.***

| Indirect effects Pathway | Bootstrap 95% CI | | | |
|---|---|---|---|---|
| | *β* | *SE* | *LL* | *UL* |
| Beliefs→Stimulation→Effortful control | −.060 | .043 | −.148 | −.010 |
| Beliefs→Stimulation→Positive affectivity | −.063 | .042 | −.144 | −.020 |
| Beliefs→Stimulation→Negative affectivity | −.016 | .022 | −.061 | .007 |
| Beliefs→Violent punishment→Effortful control | −.005 | .012 | −.024 | .015 |
| Beliefs→Violent punishment→Positive affectivity | .005 | .012 | −.018 | .021 |
| Beliefs→Violent punishment→Negative affectivity | .007 | .016 | −.016 | .035 |
| Psychological distress→Stimulation→Effortful control | −.027 | .029 | −.086 | .007 |
| Psychological distress→Stimulation→Positive affectivity | −.029 | .030 | −.085 | .006 |
| Psychological distress→Stimulation→Negative affectivity | −.007 | .013 | −.033 | .007 |
| Psychological distress→Violent punishment→Effortful control | −.036 | .025 | −.084 | −.004 |
| Psychological distress→Violent punishment→Positive affectivity | .033 | .026 | −.012 | .074 |
| Psychological distress→Violent punishment →Negative affectivity | .045 | .028 | .001 | .092 |

*Note. N* = 267. SE = standard error; CI = confidence interval; *LL* = lower limit; *UL* = upper limit.

## Discussion

The main goal of this study was to examine whether parental beliefs and psychological distress associate with parental engagement in stimulating activities and use of violent punishment and, in turn, with young children's behavioral outcomes in Bogotá and Soacha, Colombia. Understanding how beliefs and distress may impact children's outcomes through parenting behaviors is significant for global policy objectives, like the UN Sustainable Development Goals, and practice aimed at promoting children's exposure to early learning opportunities, protection from violence, and development. Doing so is particularly important in LMICs, where less than 30% of young children receive the nurturing care they require for a healthy development of biological systems and foundational skills [4].

We observed relatively low levels of endorsement of violent punishment and prevalent reported symptoms of distress in our sample of parents in Bogotá and Soacha. The low endorsement of violent punishment diverges from results of the 2018 World Values Survey, which showed that more than half of Colombians thought physical punishment is justifiable at least to some extent [63]. It may be that our non-representative sample is systematically different from a nationally representative sample, or that efforts in the last years, including the legal ban on physical punishment and the launch of the National Strategy to Prevent Violence [18] have led to reductions in caregivers' endorsement, or reports of endorsing, violent punishment. In addition, extensive evidence links socioeconomic disadvantage with increased risk for mental health challenges, which was echoed by the 2015 nationally representative Colombian Mental Health Survey [64]. Therefore, it is somewhat unsurprising to find a relatively high prevalence of at least one symptom of distress in the current sample of low-income parents in Colombia, even though we have no data to explain the causes of these symptoms.

Regarding parental behaviors, we observed relatively high levels of both stimulation and violent punishment. Although these practices were negatively correlated at the bivariate level, consistent with prior research [4], violent punishment often coexists with positive parenting behaviors. This coexistence underscores the multidimensional nature of parenting and highlights the need for interventions that both reinforce supportive practices like stimulation and address beliefs that legitimize violent discipline. Representative data from Colombia further indicate that at least 10% of infants experience physical punishment before their first birthday, with prevalence increasing as children grow older [65], aligning with our finding of a positive association between children's age and parental use of violent punishment.

Results from SEM indicate that beliefs about violent punishment predicted lower levels of children's effortful control and positive affectivity, primarily through lower levels of stimulation. While we also found a direct association between beliefs and reported use of violent punishment, these results are somewhat unexpected considering our hypothesis on content-specific relations between beliefs and behaviors. It is likely that parents who endorse violence may be more authoritative and exhibit less warmth, therefore engaging also in less stimulating activities with children [22]. At the same time, the associations between stimulation, effortful control, and positive affectivity resonate with findings from meta-analyses showing links between parental stimulation and executive function and regulatory skills, particularly for younger children [66]. Activities such as reading, storytelling, and singing have been found not only to enhance children's cognitive development but also help strengthen their attention, inhibitory control, and other self-regulatory skills in high-income and LMICs [42]. These skills are essential for developing executive functions and fostering positive emotional well-being.

Furthermore, parents' psychological distress predicted lower levels of effortful control through violent punishment. These results are aligned with Family Stress Theory [35] and the idea that psychological distress can make it harder for parents to regulate and respond in sensitive and appropriate ways to children's behaviors [24]. Associations between violent punishment, effortful control, and negative affectivity are also consistent with theoretical perspectives, as processes of observational learning, disruptions in attachment bonds, and the psychological and physiological stress caused by violent punishment is posited to impact regulatory processes and heightened negative affectivity [67]. Furthermore, given that early negative affectivity has been found to predict later conduct and externalizing problems later in childhood and adolescence [68], the current findings align with evidence from meta-analyses showing that physical and other violent punishment tends to associate more strongly with externalizing behavior problems and aggression across the life course [50].

## Implications for intervention work

Together, the current findings and previous research suggest that parental beliefs and psychological well-being play a crucial role in shaping children's development through parenting behaviors. This underscores the need for strategies that support parents in challenging beliefs that justify violent punishment and in reducing psychological distress. Parenting programs show promise in this regard, with substantial evidence demonstrating their effectiveness in enhancing parenting knowledge, self-efficacy, and positive attitudes, as well as emerging evidence on their potential to improve caregivers' mental health and well-being in LMICs [69,70]. Incorporating targeted content that challenges harmful beliefs about

violence and psychological distress into these programs could be an effective strategy for addressing key predictors of parenting behaviors and maximizing their potential impact on children's development. Future research should explore how to integrate such content in the most efficient and culturally relevant manner across diverse cultural contexts.

## Limitations and future directions

This study has important limitations that should be addressed in future research. First, the use of cross-sectional data makes it hard to discard issues of selection bias (i.e., confounding) and reversed causality. Future studies can leverage longitudinal data along more internally valid approaches like individual fixed-effects or lagged-dependent variable models to assess the extent to which observed associations may be causal. Second, our findings have limited generalizability due to the small, non-representative urban sample of parents in Bogotá and Soacha, as well as the unequal gender representation among caregivers, with most data coming from mothers and limited information on fathers or other caregivers. Replications in other parts of Colombia and in other countries are needed to assess whether these findings hold across different social and cultural contexts, and in samples that include multiple types of caregivers.

Third, all measures were parent-reported, which can lead to bias. For instance, parents may be unwilling to reveal their true beliefs and use of violent punishment, especially considering that physical punishment is legally banned in Colombia. Similarly, parents' simultaneous report of the study predictors and outcomes can cause problems of shared-method bias that can artificially inflate the observed associations. Furthermore, some indices, such as violent punishment and effortful control, showed relatively low reliability, which may result in coefficient attenuation. However, this bias would work against our hypotheses, reducing the likelihood of detecting significant associations. Thus, our results can be considered conservative, potentially representing lower-bound estimates. Fourth, some of our measures were somewhat superficial and assessed quantity (e.g., frequency of engagement in stimulation) but not necessarily quality or other key features of parental beliefs, distress, and behaviors. Future research would benefit from novel measurement approaches to understanding parents' beliefs (e.g., list experiments) and assess parent and child outcomes using different methodological approaches (e.g., direct assessment of child behavior) that enhance reliability.

Finally, our models considered static, unidirectional relations beliefs and distress → parental behaviors → child behaviors, but child behaviors may also shape parental behaviors, beliefs, and emotions. Therefore, future studies should employ longitudinal data and more dynamic modeling approaches (e.g., cross-lagged variable models) to assess reciprocity and developmental cascades between parents beliefs, distress, parenting behaviors, and child behaviors.

## Conclusion

Promoting positive parenting behaviors, such as engaging in stimulating activities and using non-violent discipline, is essential for fostering healthy child development and supporting global policy objectives like the UN Sustainable Development Goals. The current study suggests that addressing harmful attitudes toward violent punishment and alleviating parental psychological distress are critical steps in promoting positive parenting behaviors in Bogotá and Soacha, Colombia. While future research should examine the internal and external validity of these findings, the alignment with existing theory and evidence highlights the importance of policies and programs that focus on improving parental beliefs and psychological well-being as a means of enhancing young children's development in Colombia and other LMICs.

## Supporting information

**S1 Fig. Engagement in different stimulation activities.**
(DOCX)

**S2 Fig. Use of different punishment methods.**
(DOCX)

## Acknowledgments

We would like to thank Innovations for Poverty Action, the parents who participated in this study, and the staff from the ICBF, Direcciones Regionales and Centros Zonales for helping us contacting and inviting families to participate in the study. We also thank Geraldine Moreno, Maria Cristina Agudo, Óscar Pineda, Ana Serrano, Isabella García, Daniela Trujillo, and Melissa Guerra for the feedback and contributions. The statements made herein are solely the responsibility of the author(s).

## Author contributions

**Conceptualization:** Jorge Cuartas, Catalina Rey-Guerra.

**Data curation:** Jorge Cuartas.

**Formal analysis:** Jorge Cuartas, Catalina Rey-Guerra.

**Investigation:** Jorge Cuartas, Catalina Rey-Guerra, Juliana Borbón.

**Methodology:** Jorge Cuartas, Catalina Rey-Guerra, Juliana Borbón.

**Project administration:** Juliana Borbón.

**Software:** Jorge Cuartas, Catalina Rey-Guerra.

**Supervision:** Jorge Cuartas.

**Visualization:** Jorge Cuartas, Catalina Rey-Guerra.

**Writing – original draft:** Jorge Cuartas.

**Writing – review & editing:** Jorge Cuartas, Catalina Rey-Guerra, Juliana Borbón.

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
