## [Decision Letter · Decision Letter 0]

1 Sep 2025

Dear Dr. Cuartas,

Thank you for submitting your manuscript to PLOS ONE. After careful consideration, we feel that it has merit but does not fully meet PLOS ONE’s publication criteria as it currently stands. Therefore, we invite you to submit a revised version of the manuscript that addresses the points raised during the review process.

We look forward to receiving your revised manuscript.

Kind regards,

Alejandro Botero Carvajal, Ph.D

Academic Editor

PLOS ONE

Journal Requirements:

“Grants from the PanAmerican Health Association and Parenting for Lifelong Health”

“No”

5. In the online submission form you indicate that your data is not available for proprietary reasons and have provided a contact point for accessing this data. Please note that your current contact point is a co-author on this manuscript. According to our Data Policy, the contact point must not be an author on the manuscript and must be an institutional contact, ideally not an individual. Please revise your data statement to a non-author institutional point of contact, such as a data access or ethics committee, and send this to us via return email. Please also include contact information for the third party organization, and please include the full citation of where the data can be found.

“This work was supported by the Pan American Health Organization (PAHO) and The Lego Foundation. The funders had no role in the conceptualization of the study or the interpretation of results.”

“Grants from the PanAmerican Health Association and Parenting for Lifelong Health”

7. Please upload a copy of Figure 4, to which you refer in your text on page 18. If the figure is no longer to be included as part of the submission please remove all reference to it within the text

Reviewers' comments:

Reviewer's Responses to Questions

**Comments to the Author**

1. Is the manuscript technically sound, and do the data support the conclusions?

Reviewer #1: Yes

Reviewer #2: Yes

2. Has the statistical analysis been performed appropriately and rigorously?

Reviewer #1: Yes

Reviewer #2: Yes

3. Have the authors made all data underlying the findings in their manuscript fully available?

Reviewer #1: No

Reviewer #2: Yes

4. Is the manuscript presented in an intelligible fashion and written in standard English?

Reviewer #1: Yes

Reviewer #2: Yes

Reviewer #1: Thank you for the opportunity to review this manuscript that examines the influence of parenting beliefs, distress, and behaviors on child behavioral outcomes in a sample of parents in Colombia. This paper makes an important contribution to the literature, with the limitations appropriately acknowledged. I believe the findings would be of interest to readers of the PLOS ONE, however, there are several methodological and analytical clarifications that would help strengthen the manuscript if addressed and would be necessary for publication.

Methods and results:

- Given the lack of pre-registration of the analysis plan, have the authors considered sharing their survey instrument, data, analytical code, and/or analytic sample on an open science platform (with the understanding that the survey instrument was done in Spanish, I’m not suggesting that the authors translate the full instrument into English; for discussion, see Gennetian, Frank, Tamis-LeMonda, 2022)? Regarding transparency, is this the first manuscript using the data collected?

- How was the survey instrument developed? Were validated scales used? Were validated scales available in Spanish or did the authors translate them? If validated scales were translated, what was the process used for translation?

o For the CESD, can the authors provide a bit more context about why the IRB and ICBF did not approve the excluded item’s use in the population?

o The reliability for the violent punishment scale and effortful control subscale of the child behavior scales are not great. If the scales have not been previously validated in Spanish, this calls into question the use of the items as an aggregated scale. Were there items that did not hang together well with the other items that, if removed, would have improved the scale reliability? Were there mean differences (particularly for the violent punishment items) in the reporting of these behaviors? Are the findings with violent punishment being driven by one of the items?

- Were there specific inclusion/exclusion criteria for this study?

- References to the literature supporting the inclusion of the covariates missing from the introduction.

- RQ1 – Can more descriptives for the parental behaviors be provided (similar to details provided about beliefs and distress; e.g., which stimulating activities were most prevalent? Which form of violent punishment was most prevalent?)? This would provide a comprehensive answer to RQ1.

- RQ2 – Why was a measurement model conducted for beliefs and distress but not for stimulation, violent punishment, or child behavior?

- Figure 4 is missing

Typos:

- Funding disclosure lists different organizations on the cover page and title page: Lego Foundation is listed in one and Parenting for Lifelong Health is listed on another.

- Abstract: “…use of violent punishment, with in turn predicted lower effortful control…” should be “…use of violent punishment, which in turn predicted lower effortful control…”

- p. 4 last sentence of first full paragraph: “Therefore, sit remains…” should be “Therefore, it remains…”

- p. 7 first paragraph, “…or educate a child property…” should be “or educate a child properly…”

- p. 9 “…exposure to violence may children felt threaten…” I’m not sure what was meant here

- p. 11 “(Family Modality or home-based care.” is missing the closed parentheses

- p. 18 “Therefore, we Model 1 as the analytical model” is missing a verb after ‘we’

- p. 20 “…may be more authoritative and exhibit les warmth…” – where “les” should be “less”

- p. 21 “Activities such as reading, storytelling, and singing have been found not only [to] enhance children's…”

- p. 21 “…the current findings aligns with evidence…” should be “…the current findings align with evidence…”

- p. 22 the last sentence in the Implications for Intervention Work section is missing a period.

- p. 22 the first sentence in the Limitations and Future Directions section, “reserve causality” should be “reverse causality”. The third sentence in the same section should say “…given the non-representative small…”

- references – Asparouhov et al., 2013 – “Computing the stricly…” should be “Computing the strictly…”; Conger et al., 2000 is missing part of the reference (e.g., book or journal title, page numbers); There are numerous instances throughout the references where there is an extra https://doi.org in the link; the Suntheimer reference can be updated with the current year and, journal volume/issue, and page numbers; Ward et al., 2021 is missing the journal volume/issue and page number

References:

Gennetian, L. A., Frank, M. C., & Tamis-LeMonda, C. S. (2022). Open Science in developmental science. Annual Review of Developmental Psychology, 4(1), 377-397.

Reviewer #2: The study aimed to understand the influence of parental beliefs and psychological well-being on parenting behaviours and their consequential impacts on children’s behavioural outcomes. My observations are given below:

(1) The study title is okay.

(2) The abstract needs revision. A timely and pertinent study that addresses parenting beliefs, psychological distress, and child outcomes in an LMIC setting in Colombia is presented in the abstract. The objectives, sample, and important findings are stated in clear terms. However, it pays little attention to theoretical underpinnings or novelty and instead reads more like a condensed results section. The language is a little repetitive, especially when it comes to affectivity and effortful control. Transparency is decreased by the absence of methodological details (design, measurements, and analytical approach). Its impact would be increased by a more concise expression of the contribution that goes beyond context.

(3) The introduction also needs minor revision. The introduction successfully connects parenting beliefs, psychological distress, and child outcomes. It is comprehensive and contextualized within Colombian and international policy frameworks. It is overly long, though, and contains overlapping conversations that make it harder to concentrate. Although the theoretical framework is sound, the story could be condensed to better emphasize the study's distinctive contribution. Its applicability to psychology audiences would be enhanced by a deeper incorporation of psychological viewpoints beyond the focus on policy. Reducing redundancy and filling in empirical evidence gaps would increase accuracy, coherence, and the overall scholarly impact. Please enrich the review and the arguments on how parents influence the life outcomes of children based on the following studies:

Tiwari, G. K., Pandey, R., Sharma, D. N., Ray, B., Dwivedi, A., Singh, A. K., Suman, S., Singh, P., & Mishra, R. N. (2024). Understanding the protective roles of Indian joint families for children during the early phase of the COVID-19 pandemic. Journal of Qualitative Research in Health Sciences, 13(3), 1–12. http://dx.doi.org/10.22062/JQR.2024.199370.1382

Tiwari, G. K., Singh, A. K., Parihar, P., Pandey, R., Sharma, D. N., & Rai, P. K. (2023). Understanding the perceived psychological distress and health outcomes of children during COVID-19 pandemic. Educational and Developmental Psychologist, 40(1), 103–114. https://doi.org/10.1080/20590776.2021.1899749

Tiwari, G. K., Tiwari, R. P., Pandey, R., Ray, B., Dwivedi, A., Sharma, D. N., Singh, P., Tiwari, A. K., & Singh, A. K. (2024). Perceived Life Outcomes of Indian Children During the Early Phase of the COVID-19 Lockdown: The Protective Roles of Joint and Nuclear Families. Journal of Research and Health, 14(1), 43–54. https://doi.org/10.32598/JRH.14.1.1992.4

(4) The method again needs minor amendments. Positionality, ethics, sampling, measurements, and analytical techniques are all integrated in the comprehensive methods section. Although positionality and transparency are added to increase credibility, social desirability bias is a concern due to the use of self-report data. Additional justification for item removal and validity implications are warranted for the modified CES-D-10. Generalizability is limited by the unequal gender representation of caregivers (mothers are overwhelmingly dominant). SEM is suitable, but it would be better to address the presumptions and possible drawbacks of clustered sampling. Methodological rigor would be further strengthened by providing a clearer justification for reliability thresholds and measure selection.

(5) The results may be improved. The results section is thorough and rationally organised, successfully tying together measurement validation, structural modelling, and descriptive findings. However, readers who are not familiar with SEM may find the presentation overwhelming due to its denseness and excessive statistical detail. Accessibility would be improved by placing more focus on practical interpretation, such as the implications of high distress prevalence for parenting and child outcomes. Furthermore, there is little discussion of non-significant results, which runs the risk of favouring positive findings over others. Coherence would be strengthened and more significant psychological insights would be supported by a clearer integration of descriptive and inferential results.

(6) The discussion needs minor improvements. It is thorough, theoretically supported, and skilfully connects research results to more general policy and developmental ramifications. Nevertheless, it occasionally restates findings without offering enough critical synthesis, which reduces the depth of interpretation. Clarity would be enhanced by a stronger focus on elucidating surprising results, especially the coexistence of stimulation and harsh punishment. Although the section acknowledges its limitations, it could do a better job of balancing its strengths and limitations. Furthermore, while encouraging, the implications for interventions are still rather broad; more culturally specific approaches would increase the discussion's usefulness.

**Do you want your identity to be public for this peer review?** For information about this choice, including consent withdrawal, please see our Privacy Policy

Reviewer #1: No

Reviewer #2: **Yes: ** Gyanesh Kumar Tiwari

---

## [Author Response · Author response to Decision Letter 1]

7 Oct 2025

Response to the Editor

Thank you very much for the comments and the opportunity to revise our manuscript for publication in PLOS ONE. We have used the reviewers’ thoughtful and constructive feedback to make substantial edits to the manuscript, which we believe have greatly improved the paper. Updates to the manuscript are detailed below.

Response to Referee #1

1. Thank you for the opportunity to review this manuscript that examines the influence of parenting beliefs, distress, and behaviors on child behavioral outcomes in a sample of parents in Colombia. This paper makes an important contribution to the literature, with the limitations appropriately acknowledged. I believe the findings would be of interest to readers of the PLOS ONE, however, there are several methodological and analytical clarifications that would help strengthen the manuscript if addressed and would be necessary for publication.

Response: Thank you very much for your positive and constructive feedback

2. Methods and results: Given the lack of pre-registration of the analysis plan, have the authors considered sharing their survey instrument, data, analytical code, and/or analytic sample on an open science platform (with the understanding that the survey instrument was done in Spanish, I’m not suggesting that the authors translate the full instrument into English; for discussion, see Gennetian, Frank, Tamis-LeMonda, 2022)?

Response: In response, we have uploaded the suggested materials to the Open Science Framework repository (https://osf.io/xv9ma/?view_only=0a439c7a76b648418ea5a727c3bc7c63) and included these details in the manuscript as follows (p. 11): “The data, measurement instruments, and analytic code for this study are available through our project’s Open Science Framework repository at https://osf.io/xv9ma/?view_only=0a439c7a76b648418ea5a727c3bc7c63”

3. How was the survey instrument developed? Were validated scales used? Were validated scales available in Spanish or did the authors translate them? If validated scales were translated, what was the process used for translation?

Response: Thank you for this important comment. Our instrument combined both established scales and newly developed items. For example, we drew on items from the Family Care Indicators (available in Spanish through the Multiple Indicator Cluster Survey), as well as the Spanish versions of the CES-D-10 and the Infant and Early Childhood Behavior Questionnaires. As described in the Methods section, these instruments have been previously used in Colombia (e.g., Attanasio et al., 2014). At the same time, we developed new indices, such as the Attitudes Toward Violence Index, because we were unable to identify existing measures with adequate psychometric properties for the Colombian context.

Regarding validity, we relied on instruments widely used in low- and middle-income countries, many with demonstrated psychometric adequacy in Colombia. Importantly, as validity does not reside in the scale itself but rather ‘refers to the degree to which evidence and theory support the interpretations of test scores for proposed uses of tests’ (see APA Standards), we conducted psychometric analyses for the current population, particularly for scales not previously assessed in Colombia (e.g., attitudes)

4. For the CESD, can the authors provide a bit more context about why the IRB and ICBF did not approve the excluded item’s use in the population?

Response: In response, to edited the text to clarify as follows: “We removed the item I could not "get going" because the ICBF did not approve its use in populations receiving their services, stating that it was misaligned with their policy framework” (p. 13)

5. The reliability for the violent punishment scale and effortful control subscale of the child behavior scales are not great. If the scales have not been previously validated in Spanish, this calls into question the use of the items as an aggregated scale. Were there items that did not hang together well with the other items that, if removed, would have improved the scale reliability? Were there mean differences (particularly for the violent punishment items) in the reporting of these behaviors? Are the findings with violent punishment being driven by one of the items?

Response: We appreciate this comment. In response, we have added additional information to the Limitations section regarding the lower reliability of these indices and its implications for interpreting our results, as follows (p. 23): “Furthermore, some indices, such as violent punishment and effortful control, showed relatively low reliability, which may result in coefficient attenuation. However, this bias would work against our hypotheses, reducing the likelihood of detecting significant associations. Thus, our results can be considered conservative, potentially representing lower-bound estimates.”

Indeed, while we agree that the reliability of these indices is lower than that of other scales, it is important to note that lower reliability primarily increases error and may attenuate coefficients, making it less likely that statistically significant associations are detected. Consequently, even in the presence of substantial measurement error, our findings likely represent conservative or “lower-bound” estimates of the true associations between the variables of interest.

Regardless of coefficient attenuation, the reliability of the violent punishment index is consistent with prior studies using similar items. The observed reliability is further affected by the small sample size and the limited number of items, both of which tend to reduce estimates from Cronbach’s alpha. Importantly, the violent punishment items are intended to calculate a count index of the number of different forms of violent punishment used, rather than to estimate a latent factor such as aggressiveness. We also provide additional evidence on the reliability of this index: no single item drives lower reliability, and all items show adequate correlations in the expected direction.

Average

Item-test Item-rest interitem

Item Obs Sign correlation correlation covariance alpha

d_shook 267 + 0.28 0.16 0.05 0.60

d_names 267 + 0.26 0.12 0.05 0.60

d_ign 267 + 0.62 0.27 0.04 0.60

d_thr 267 + 0.68 0.51 0.03 0.49

d_spank 267 + 0.63 0.33 0.04 0.56

d_obj 267 + 0.57 0.47 0.04 0.54

d_yell 267 + 0.70 0.49 0.03 0.48

Test scale 0.04 0.60

For effortful control, reliability is relatively adequate and aligns with evidence from other instruments, such as the International Development and Early Learning Assessment (IDELA), for which executive function, closely related to effortful control, often exhibits lower reliability than other domains (e.g., Cuartas et al., 2023, Developmental Science).

6. Were there specific inclusion/exclusion criteria for this study?

Response: Yes, and in response, we have added a sentence to the Procedures and Sample subsection clarifying that eligibility criteria included participation in Modalidad Familiar services and having a child younger than five, as follows (p. 11-12): “We held meetings with coordinators of all Modalidad Familiar Service Units (UDS) in the targeted areas to identify which units could participate based on availability. Seventeen UDS were selected, and all caregivers in these units with children under five were eligible and invited to provide written informed consent”

7. References to the literature supporting the inclusion of the covariates missing from the introduction.

Response: Thank you for this suggestion. In response, we have added a mention, justification, and supporting references for the inclusion of key covariates, as follows (p. 5): “An ecological perspective on parenting suggests that (a) parent characteristics (e.g., beliefs and emotions), (b) child characteristics (e.g., age and gender), and (c) the household and family environment (e.g., parent education, employment, household composition, and wealth) interact to shape parenting and child development (Bornstein, 2015). At the same time, more distal factors such as parental employment or household wealth likely exert their influence primarily through proximal predictors, including parents’ cognitions and emotions (Bornstein, 2016; Klahr & Burt, 2014; Taraban & Shaw, 2018).”

8. RQ1 – Can more descriptives for the parental behaviors be provided (similar to details provided about beliefs and distress; e.g., which stimulating activities were most prevalent? Which form of violent punishment was most prevalent?)? This would provide a comprehensive answer to RQ1.

Response: We appreciate this helpful suggestion and have responded to it by including detailed descriptive statistics for stimulation and punishment in the supplemental file. We also refer readers to it in the text, as follows (p. 17): “Regarding parental behaviors, the data indicated relatively high engagement in stimulating activities (M = 2.99 out of 4, SD = 0.51), alongside a prevalent use of violent punishment (M = 1.16 out of 4, SD = 0.26; See Figures S1 and S2 in the supplemental file).”

9. RQ2 – Why was a measurement model conducted for beliefs and distress but not for stimulation, violent punishment, or child behavior?

Response: Thanks. We conducted measurement models for beliefs and distress because these scales are intended to capture latent (i.e., unobserved) constructs and do not have established scoring procedures; the models test whether they reliably measure key aspects of parental cognitions and well-being. In contrast, stimulation and violent punishment are count indices designed to assess the number of different activities or punishment methods parents used, and thus do not measure latent constructs. For child behavior, we followed the developer’s scoring procedures, so no measurement model was conducted to estimate a latent factor.

10. Figure 4 is missing

Response: the revised document including all Figure now includes Figure 4.

11. Typos: Funding disclosure lists different organizations on the cover page and title page: Lego Foundation is listed in one and Parenting for Lifelong Health is listed on another.

Abstract: “…use of violent punishment, with in turn predicted lower effortful control…” should be “…use of violent punishment, which in turn predicted lower effortful control…”

p. 4 last sentence of first full paragraph: “Therefore, sit remains…” should be “Therefore, it remains…”

p. 7 first paragraph, “…or educate a child property…” should be “or educate a child properly…”

p. 9 “…exposure to violence may children felt threaten…” I’m not sure what was meant here

p. 11 “(Family Modality or home-based care.” is missing the closed parentheses

p. 18 “Therefore, we Model 1 as the analytical model” is missing a verb after ‘we’

p. 20 “…may be more authoritative and exhibit les warmth…” – where “les” should be “less”

p. 21 “Activities such as reading, storytelling, and singing have been found not only [to] enhance children's…”

p. 21 “…the current findings aligns with evidence…” should be “…the current findings align with evidence…”

p. 22 the last sentence in the Implications for Intervention Work section is missing a period.

p. 22 the first sentence in the Limitations and Future Directions section, “reserve causality” should be “reverse causality”. The third sentence in the same section should say “…given the non-representative small…”

Response: We sincerely appreciate your dedication and thorough review. We have corrected all the identified typos and carefully revised the text overall to ensure it is free of additional errors.

12. references – Asparouhov et al., 2013 – “Computing the stricly…” should be “Computing the strictly…”; Conger et al., 2000 is missing part of the reference (e.g., book or journal title, page numbers); There are numerous instances throughout the references where there is an extra https://doi.org in the link; the Suntheimer reference can be updated with the current year and, journal volume/issue, and page numbers; Ward et al., 2021 is missing the journal volume/issue and page number

Response: Thank you for these detailed suggestions. We have included all the suggested edits.

Response to Referee #2

13. The study aimed to understand the influence of parental beliefs and psychological well-being on parenting behaviours and their consequential impacts on children’s behavioural outcomes. My observations are given below:

Response: Thank you for your thorough review and helpful feedback

14. The study title is okay.

Response: Thank you

15. The abstract needs revision. A timely and pertinent study that addresses parenting beliefs, psychological distress, and child outcomes in an LMIC setting in Colombia is presented in the abstract. The objectives, sample, and important findings are stated in clear terms. However, it pays little attention to theoretical underpinnings or novelty and instead reads more like a condensed results section. The language is a little repetitive, especially when it comes to affectivity and effortful control. Transparency is decreased by the absence of methodological details (design, measurements, and analytical approach). Its impact would be increased by a more concise expression of the contribution that goes beyond context.

Response: Thank you very much for these excellent suggestions. We have thoroughly edited the abstract following your guidance.

16. The introduction also needs minor revision. The introduction successfully connects parenting beliefs, psychological distress, and child outcomes. It is comprehensive and contextualized within Colombian and international policy frameworks. It is overly long, though, and contains overlapping conversations that make it harder to concentrate. Although the theoretical framework is sound, the story could be condensed to better emphasize the study's distinctive contribution. Its applicability to psychology audiences would be enhanced by a deeper incorporation of psychological viewpoints beyond the focus on policy. Reducing redundancy and filling in empirical evidence gaps would increase accuracy, coherence, and the overall scholarly impact.

Response: In response, we have edited the introduction to streamline some content and strengthen the clarity of our specific contributions. We have also incorporated multiple major psychological theories and viewpoints across the introduction, including, among others, perspectives by Ajzen, Bandura, Bornfenbrenner, Bornstein, Conger, Gershoff, Lansford, and McLaughlin, among others.

17. The method again needs minor amendments. Positionality, ethics, sampling, measurements, and analytical techniques are all integrated in the comprehensive methods section. Although positionality and transparency are added to increase credibility, social desirability bias is a concern due to the use of self-report data. Additional justification for item removal and validity implications are warranted for the modified CES-D-10. Generalizability is limited by the unequal gender representation of caregivers (mothers are overwhelmingly dominant). SEM is suitable, but it would be better to address the presumptions and possible drawbacks of clustered sampling.

Response: Thank you for these comments. We made several edits to the manuscript in response. First, we have included in the Limitations section a conversation about social desirability bias, as follows (p. 23): “Third, all measures were parent-reported, which can lead to bias. For instance, parents may be unwilling to reveal their true beliefs and use of violent punishment, especially considering that physical punishment is legally banned in Colombia. Similarly, parents’ simultaneous report of the study predictors and outcomes can cause problems of shared-method bias that can artificially inflate the observed associations.”

Second, We have also edited the text to provide further information on the exclusion of one item of the CES-D-10, as follows: “We removed the item I could not "get going" because the ICBF did not approve its use in populations receiving their services, stating that it was misaligned with their policy framework” (p. 13)

Third, we have included additional information about the unequal gender representation in the Limitations section, as follows (p. 22): Second, our findings have

---

## [Decision Letter · Decision Letter 1]

22 Oct 2025

Parenting Beliefs and Psychological Distress -> Stimulation and Punishment-> Young Children’s Behavior: A Descriptive Study in Colombia

PONE-D-25-30848R1

Dear Dr. Cuartas,

We’re pleased to inform you that your manuscript has been judged scientifically suitable for publication and will be formally accepted for publication once it meets all outstanding technical requirements.

Kind regards,

Alejandro Botero Carvajal, Ph.D

Academic Editor

PLOS ONE

Additional Editor Comments (optional):

Reviewers' comments:

Reviewer's Responses to Questions

**Comments to the Author**

Reviewer #2: All comments have been addressed

2. Is the manuscript technically sound, and do the data support the conclusions?

Reviewer #2: Yes

3. Has the statistical analysis been performed appropriately and rigorously?

Reviewer #2: Yes

4. Have the authors made all data underlying the findings in their manuscript fully available?

Reviewer #2: Yes

5. Is the manuscript presented in an intelligible fashion and written in standard English?

Reviewer #2: (No Response)

Reviewer #2: The revised manuscript is now more readable and technically sound. It is now more suitable for the journal and now thus may contribute the the rea of scietific knowledge.

Well done. No furtner amendments are needed.

**Do you want your identity to be public for this peer review?** For information about this choice, including consent withdrawal, please see our Privacy Policy

Reviewer #2: **Yes: ** Gyanesh Kumar Tiwari

---

## [Editor Report · Acceptance letter]

PONE-D-25-30848R1

PLOS ONE

Dear Dr. Cuartas,

I'm pleased to inform you that your manuscript has been deemed suitable for publication in PLOS ONE. Congratulations! Your manuscript is now being handed over to our production team.

Kind regards,

on behalf of

Dr. Alejandro Botero Carvajal

Academic Editor

PLOS ONE